# Using the Volta phase plate with defocus for cryo-EM single particle analysis

**Radostin Danev[1]\*, Dimitry Tegunov[2], Wolfgang Baumeister[1]**

[1]Department of Molecular Structural Biology, Max Planck Institute of Biochemistry, Martinsried, Germany; [2]Department of Molecular Biology, Max Planck Institute for Biophysical Chemistry, Göttingen, Germany

**Abstract** Previously, we reported an in-focus data acquisition method for cryo-EM single-particle analysis with the Volta phase plate (Danev and Baumeister, 2016). Here, we extend the technique to include a small amount of defocus which enables contrast transfer function measurement and correction. This hybrid approach simplifies the experiment and increases the data acquisition speed. It also removes the resolution limit inherent to the in-focus method thus allowing 3D reconstructions with resolutions better than 3 Å.

## Introduction

The Volta phase plate (VPP) is a device which enables in-focus phase contrast in a transmission electron microscope (TEM) (*Danev et al., 2014*). It is the first phase-plate-type device to reach near-atomic resolutions in single particle cryo-EM (*Chua et al., 2016*; *Danev and Baumeister, 2016*; *Khoshouei et al., 2016a*, *2016b*). Nowadays such resolutions are not uncommon, but there are still challenges related mainly to intrinsic properties of the target structures. Small, heterogeneous and/or flexible molecules are generally difficult to solve by the traditional defocus phase contrast (DPC) method (*Cheng et al., 2015*). The VPP improves image contrast and thus could help with solving the structures of such 'difficult' samples. Aside from the improved contrast, one of the main questions about the VPP in the cryo-EM community has been: does it in any way limit the achievable resolution as compared to DPC?

In our previous report (*Danev and Baumeister, 2016*) we presented an in-focus cryo-EM approach with the VPP. The in-focus method, although ideal from a theoretical point of view, has a few practical disadvantages and limitations. It requires very accurate focusing which complicates and slows down the data acquisition. Furthermore, the achievable resolution is limited by the spherical aberration of the objective lens to ~3 Å. At the time of our initial report the 3D reconstruction software did not yet support VPP phase shift (PS) in the contrast transfer function (CTF). In the meantime, such support was implemented which enabled the present work. Here, we present a hybrid approach which combines the VPP with a small amount of defocus. It simplifies the data acquisition to a level similar to DPC and solves the resolution limit issue of the in-focus approach by enabling CTF fitting and correction.

## Results and discussion

In our report on in-focus cryo-EM with the VPP we used *Thermoplasma acidophilum* 20S proteasome as a test sample (*Danev and Baumeister, 2016*). Here, we used the same sample for consistency and continuation. The defocused VPP dataset is illustrated in *Figure 1*. Acquiring images with defocus is easier and faster because the defocus and astigmatism can be fitted and corrected a posteriori. The VPP provides low frequency contrast which allows the use of defocus values smaller than

**\*For correspondence:** danev@biochem.mpg.de

**Figure 1.** Volta phase plate with defocus cryo-EM dataset of 20S proteasome. (**A**) Representative image of 20S proteasomes in ice, defocus 500 nm. (**B**) Power spectrum of the image in (**A**) showing contrast transfer function rings (Thon rings). To enhance the visibility of Thon rings, the power spectrum was calculated as the sum of the power spectra of individual movie frames (*McMullan et al., 2015*). (**C**) Phase shift history throughout the dataset. The phase shift gradually increases until the phase plate is moved to a new position where it suddenly drops and starts to raise again. (**D**) Histogram illustrating the phase shift distribution. (**E**) Defocus history throughout the dataset. The target defocus was changed after ~200 images from 500 nm to 300 nm. (**F**) Histograms illustrating the defocus distributions. Scale bar: 50 nm.

those typically used in DPC. Because of the low defocus the images (*Figure 1A*) look similar to in-focus images (Figure 4A in *Danev and Baumeister, 2016*). The power spectrum in *Figure 1B*, how-ever, clearly shows the effect of defocus with characteristic CTF rings (Thon rings). Fitting the CTFs of VPP data requires an additional PS parameter (*Rohou and Grigorieff, 2015*). Such fits provide a quantitative measure of the behavior of the VPP. *Figure 1C* shows a plot of the PS history through-out the dataset. The VPP was advanced to a new position every 1.5 hr (every ~40 images, total dose on the VPP ~50 nC). After each advance the PS drops abruptly to a low value (<0.2 $\pi$) and starts to

gradually build up again. *Figure 1D* contains a histogram of the measured PS. The distribution has a maximum at ~0.6 π with a relatively small number of micrographs exhibiting low (<0.2 π) or high (>0.8 π) PS. The evolving PS of the VPP is an advantage for single particle analysis because it moves the positions of the CTF zeros thus mitigating the need to vary the defocus, which is necessary with DPC. With the VPP, datasets can be collected with a single, low defocus value.

In this work (*Figure 1C*) as well as in the previous report (Figure 3 in *Danev and Baumeister, 2016*) the VPP exhibited more PS than in the original VPP paper (*Figure 1C* in *Danev et al., 2014*). In addition, there is some variation in the PS magnitude between different areas on the phase plate (*Figure 1C*). In practice, it is advantageous to have a VPP with slower PS development because this allows collection of more images at each position. The heater used in the original VPP paper (*Danev et al., 2014*) was a pre-production prototype and the phase plate was homemade thus the production versions used in this work seem to exhibit a faster PS development. The variation of the PS across VPP areas could indicate a local variation in the quality of the amorphous carbon film. Overall, those factors do not prevent the collection of high quality data, as illustrated by the results presented here, but there is definitely room for improvement and we hope that manufacturers will take those observations into account when they develop future versions of the hardware.

The history of the measured defocus is plotted in *Figure 1E*. Approximately halfway through the dataset acquisition we changed the target defocus from 500 nm to 300 nm to evaluate the performance at different defocus values. The measured defocus has periodic oscillations, with ~16 image period, probably caused by local variations in the slant of the support film (waviness), which can introduce a defocus difference between the focusing and acquisition positions. Histograms of the measured defocus values are shown in *Figure 1F*. The distribution of the 300 nm target defocus data is wider than the 500 nm one but this seems to be caused by the systematic defocus oscillations and not by random focusing errors (*Figure 1E*). The 300 nm target defocus had a practical disadvantage in that fitting the CTFs of micrographs with defocus <300 nm was difficult because the CTF has fewer rings and their period is similar to power spectrum features, such as the amorphous ice ring at ~3.7 Å. In practice, the 500 nm defocus was more robust and easier to process.

High PS (>0.8 π) is undesirable because it causes CTF artifacts and thus reduces the quality of the data. *Figure 2A* shows examples of images with different amounts of PS. The image on the left has a low PS (0.1 π) and consequently lower contrast. The middle image is close to the optimal PS of 0.5 π and has good contrast and the best appearance. The image on the right was acquired with a high PS of 0.9 π and looks 'blurry'. The blurriness is caused by an extra CTF maximum at very low spatial frequencies illustrated in *Figure 2B*. In practice, it is difficult to predict the exact shape or size of the central spot of the VPP. Factors such as specimen charging, specimen thickness and beam drift could cause the central beam to change its shape or move on the phase plate and thus modify the central spot. We approximated the central spot of the VPP with a Gaussian function which has only one parameter – the size of the spot. This is a simple model which, although not accurate in every case, provides a satisfactory simulation of a rather unpredictable function. The bottom plot in *Figure 2B* shows the approximated relative PS profile of the VPP. The top plot shows CTFs for various saturation PS values. The CTFs were calculated using the formula:

$$|CTF(k)| = \left| \sin\left[ -\varphi\left(1 - e^{-\frac{k^2}{2s^2}}\right) + \pi\left(-\Delta z\,\lambda\,k^2 + \frac{1}{2}C_S\,\lambda^3\,k^4\right)\right]\right| \tag{1}$$

where $\varphi$ is the phase shift of the VPP in radians, $k$ is the spatial frequency, $s$ is the radius of the VPP spot in reciprocal space (0.05 1/nm), $\triangle z$ is the defocus (500 nm), $\lambda$ is the electron wavelength (0.002 nm) and $C_S$ is the spherical aberration coefficient ($2.6 \times 10^6$ nm). In the case of an ideal phase plate the VPP spot is infinitely small, i.e. $s = 0$. Up to a PS of 0.5 π (dotted lines) the CTFs behave well in a sense that they rise gradually in accordance with the PS profile. For high PSs (red line) the CTF rises quickly and passes through a maximum at the point where the PS crosses 0.5 π on its way to the saturation value. This peak at very low spatial frequencies and the shift of the first CTF zero towards the low frequencies are the reasons for the blurry appearance of the high PS image in *Figure 2A*. The CTF model used in data processing is that of an ideal phase plate with a delta-function-like central spot (black dashed line). The large deviation between the practical and the theoretical CTFs (pink area in *Figure 2B*) means that high PS images are not handled optimally during processing.

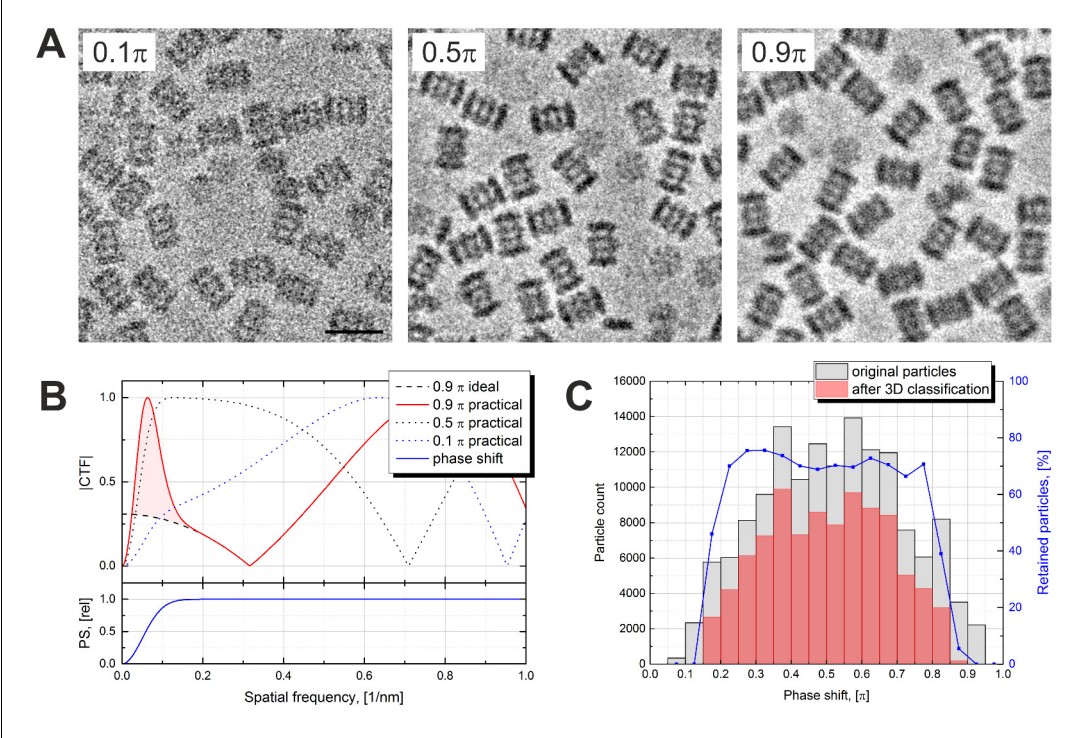

**Figure 2.** Effects of the Volta phase plate phase shift on the image appearance and the contrast transfer function. (**A**) Examples of images at low (0.1 π), optimal (0.5 π) and high (0.9 π) phase shifts. (**B**) Simulated contrast transfer functions (CTF) at 500 nm defocus and different phase shifts (top). Relative phase shift (PS) profile of the Volta phase plate (bottom). The black dashed line represents a CTF of an ideal (delta function) phase plate with 0.9 π phase shift. (**C**) Phase shift histograms before (gray) and after (red) 3D classification of the particles. Particles with low (<0.2 π) and high (>0.8 π) phase shifts were predominantly rejected. The blue line (right vertical axis) is the particle retention (after vs before 3D classification). Scale bar: 20 nm.

*Figure 2C* shows PS histograms of the particles before and after 3D classification. Particles with low (<0.2 π) and high (>0.8 π) PSs were predominantly rejected indicating that such particles had low correspondence with the reference. There was no correlation between defocus and particle retention. The first few images after moving the VPP to a new position have a low PS which quickly develops (*Figure 3* in *Danev and Baumeister, 2016*). In those images there may be noticeable PS evolution throughout the multiframe movie. A CTF fit taking into account such evolution is possible, but in general, throwing away such images has little effect on the data acquisition throughput. For high PS images, the CTF model can, in principle, be modified to include the central VPP spot which should improve their handling. However, it is best to avoid both low and high PS images which could be achieved through an improved data acquisition strategy. An optimal solution would be to incorporate PS monitoring in the automated data acquisition software through on-the-fly CTF fitting of the acquired images. Once the PS reaches a preset value, e.g. 0.8 π, the VPP is advanced to the next position and pre-irradiated for a given time to build up a desired initial PS, e.g. 0.2 π, before continuing the acquisition. The dataset presented here was acquired with less frequent changes of the VPP position and without throwing away the first few images in order to explore the effect of PS on the quality of the data.

We first processed the data through the standard Relion workflow (*Scheres, 2012*, *2014*). The results are presented in *Figure 3*. The reconstruction reached a resolution of 2.4 Å according to the gold-standard Fourier shell correlation (FSC) 0.143 criterion (*Figure 3C*). The visibility of primary structure features, such as the holes in the tyrosine and phenylalanine rings (*Figure 3B*), is in accordance with that estimate. The local resolution varies between 2.2 Å and 2.6 Å (*Figure 3A*). The significant improvement in resolution compared to our previous in-focus dataset, at 3.2 Å (*Danev and Baumeister, 2016*), confirms the effectiveness of the CTF correction. *Figure 3E* shows a plot of the resolution versus number of particles calculated using random particle subsets. The resolution

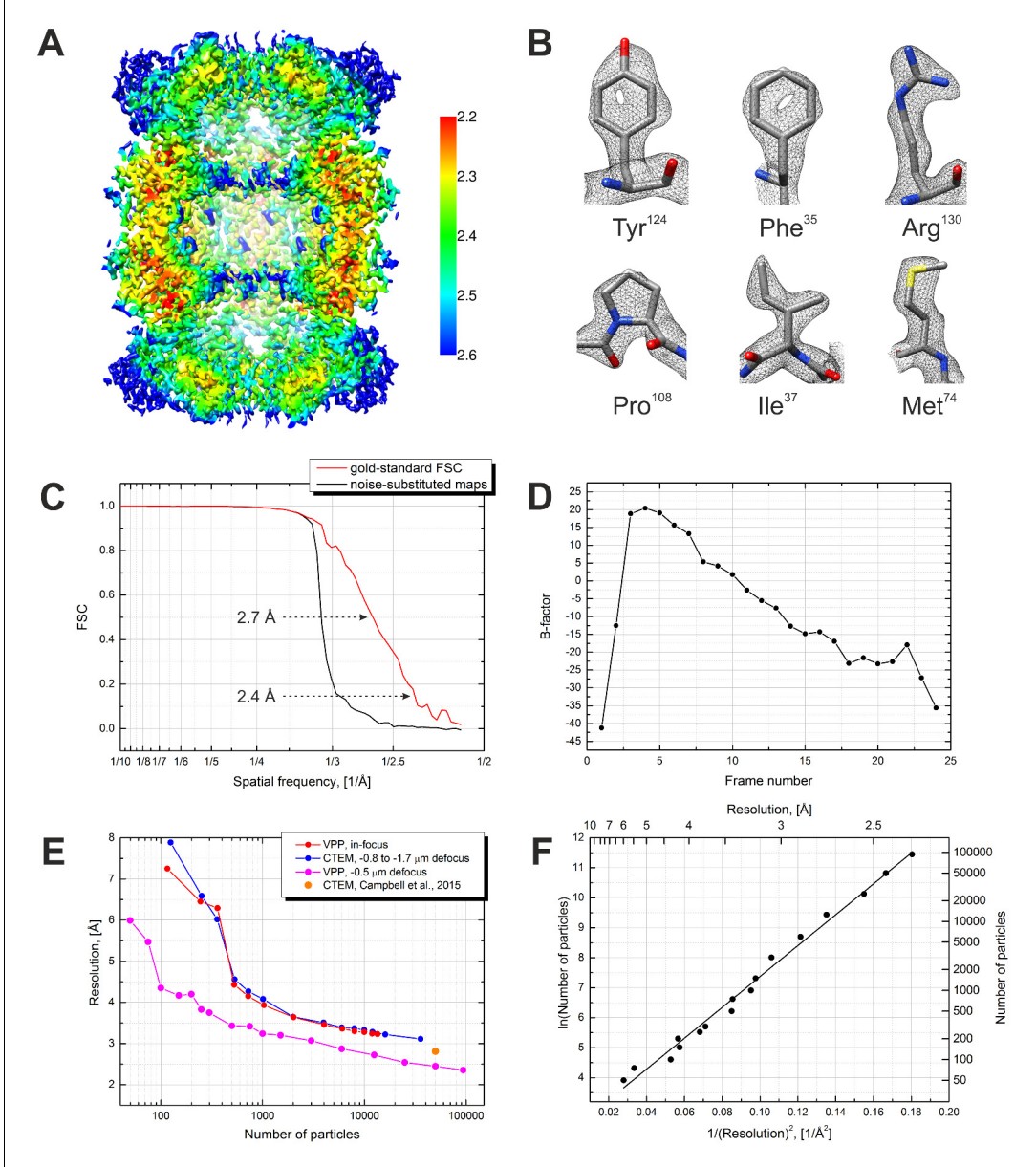

**Figure 3.** Results from the 3D reconstruction of the 20S proteasome dataset with Relion particle polishing. (A) Cross-section of the 3D map colored according to the local resolution. (B) Examples of sidechain details. (C) Fourier shell correlation (FSC) plots indicating a resolution of 2.4 Å according to the gold-standard FSC = 0.143 criterion. (D) Per-frame B-factors calculated during the particle polishing. (E) Resolution as a function of the number of particles measured using random particle subsets. Also shown is a data point from *Campbell et al. (2015)*. (F) Same data as in (E) but with logarithmic and squared reciprocal axes. The slope of the linear fit indicates an overall B-factor of 103 Å$^2$.

improvement is consistent throughout the particle number range with a significant extension towards low particle numbers. The improvement is also a result of the fact that the defocused data-set was collected with a smaller pixel size (1.054 Å vs 1.35 Å previously) and in super-resolution mode of the K2 camera. *Figure 3F* contains a plot of the logarithm of the number of particles vs the squared reciprocal resolution, with a linear fit. The B-factor calculated as twice the slope of the fit (*Rosenthal and Henderson, 2003*) was 103 Å$^2$. In this work we did not collect a DPC dataset because our main goal was to explore the capabilities of the VPP. We expect that with an 'easy' specimen, such as the 20S proteasome, DPC would have produced very similar results to those of the VPP.

We also tried a simplified data processing workflow based on the MotionCor2 software (*Zheng et al., 2016*). It performs movie frame alignment with local motion tracking and dose-weighting. Such preprocessing provides similar performance gains to particle polishing in Relion (*Scheres, 2014*) but is more efficient in terms of computation and storage space. The results was, for all practical purposes, virtually identical to that from particle polishing both in terms of resolution (2.42 Å MotionCor2 vs 2.36 Å polishing) and B-factor (85 Å$^2$ MotionCor2 vs 74 Å$^2$ polishing). Because of its simplicity, speed and comparable performance, nowadays we use the MotionCor2 workflow for most of our projects.

In summary, the use of VPP with defocus simplifies the data acquisition and improves the acquisition speed. The evolving PS of the VPP allows the use of a constant, low target defocus. In our experience, 500 nm is a good practical value because it ensures reliable CTF fits with some margin for random or systematic focusing errors. Future modifications of the data acquisition software, to include on-the-fly CTF fitting, PS monitoring and VPP pre-irradiation, would improve the performance by preventing the collection of low and high PS images. Here, we demonstrated that the VPP is not limited in resolution and matches the state of the art of DPC for 'easy' samples, such as the 20S proteasome. The real performance advantage of the VPP needs further exploration using 'difficult' samples, such as small, flexible and/or heterogeneous samples.

## Materials and methods

### Data acquisition

Cryo-samples were prepared as described previously (*Danev and Baumeister, 2016*). The data was collected on an FEI Ttian Krios (FEI, Hillsboro, OR) electron microscope operated at 300 kV and equipped with a Gatan GIF Quantum energy filter and a Gatan K2 Summit direct detection camera (Gatan, Pleasanton, CA). The acquisition conditions were as follows: EFTEM Nanoprobe mode, magnification x47,000, 50 µm C2 aperture, spot size 6, beam diameter 1.2 µm, beam current 0.105 nA, zero-loss imaging with 20 eV slit, K2 Summit in super-resolution mode, physical pixel size 1.06 Å, total dose 39 e$^-$/Å$^2$, dose rate on the detector 3.6 e$^-$/pixel/s, exposure time 12 s, 24 frames 0.5 s each. The data was acquired automatically with SerialEM software (*Mastronarde, 2005*). Focusing was performed next to each hole with three image focusing (drift protection), 10 mrad beam tilt, zero defocus offset and taking into account the effect of spherical aberration (target defocus = desired defocus + 270 nm; e.g. for 500 nm underfocus the target defocus was set to −230 nm). The VPP was advanced to a new position every 1.5 hr (every ~40 images, total dose on the VPP ~50 nC). The data acquisition speed was ~27 images/hour, comparable to that of the previously collected in-focus dataset (*Danev and Baumeister, 2016*). The throughput did not improve, despite simplified focusing, because of the use of super-resolution mode of the K2 camera which added ~50 s to the time required to process and save the movie frames. The movies were saved as dark-subtracted but non-gain-normalized LZW compressed TIFF files which significantly reduced their size (~220 MB/movie). The gain reference was saved separately.

### Data processing workflow with relion particle polishing

The dataset consisted of 468 micrographs, 41 (9%) of which were rejected after visual inspection, leaving 427 micrographs for processing. The super-resolution movies were first converted to MRC format with the *newstack* from IMOD (*Mastronarde and Held, 2016*), which increased their size to 5.1 GB/movie. Magnification anisotropy correction (measured to be 1.2%) and gain normalization were performed with *mag_distortion_correct* (*Grant and Grigorieff, 2015*). The movie frames were aligned with *unblur* (*Campbell et al., 2012*). Particles were picked by template matching in Gautomatch (http://www.mrc-lmb.cam.ac.uk/kzhang/). The CTFs were fitted with a homemade program in MATLAB (MathWorks, Natick, MA). The rest of the processing was performed in the GPU-accelerated beta version of Relion 2.0 (*Kimanius et al., 2016*). Particles were extracted from the aligned super-resolution micrographs and movies with 2x downsampling, resulting in a pixels size of 1.054 Å, matching the physical pixel size of the camera. The box size after downsampling was 180 pixels. The initial dataset of 145,870 particles was subjected to a 3D refinement with D7 symmetry which reached a resolution of 2.6 Å with a B-factor of 111 Å$^2$. The output of the refinement was used to 'polish' the particles with the following parameters: running average three frames, stddev

on translations one pixel, stddev on particle distance 100 pixels, B-factor highres-limit 2 Å, B-factor lowres-limit 7 Å. The polished dataset was 3D classified into five classes over several steps with finer and finer angular and translational sampling and E-step resolution limit of 12 Å. The best class, containing 93,596 particles (64% of the dataset), was 3D refined producing the final reconstruction with a resolution of 2.4 Å and a B-factor of 74 $Å^2$. The other 3D classes refined to much lower resolutions with one of them showing some structure deformation in the form of separation between the top and bottom halves of the 20S. This was probably due to an incomplete assembly of some particles rather than being a valid functional state. To produce the resolution vs particle number data, subsets of random particles were extracted from the final dataset and 3D refined separately with a 60 Å low-pass filtered initial reference. The local resolution distributions were calculated with *blocres* from Bsoft (*Heymann and Belnap, 2007*).

## Data processing workflow with MotionCor2

For the MotioCor2 (*Zheng et al., 2016*) based reconstruction, the movies were gain-normalized and aligned using 9 × 9 patches, 2x downsampling, first two frames discarded and dose-weighting with 1.0 $e^-$/ $Å^2$/frame which was ~1.5 times lower than the measured frame dose of 1.625 $e^-$/ $Å^2$/frame. The reduced dose-weighting parameter was determined empirically by trying different values and selecting the value which produced the highest resolution reconstructions. The CTFs were fitted on the non-dose-weighted micrographs with ctffind4 (*Rohou and Grigorieff, 2015*) version 4.1.5 using the following parameters: spherical aberration 2.62 mm, minimum resolution 20 Å, maximum resolution 3 Å, minimum defocus 3000 Å, maximum defocus 7000 Å, minimum phase shift 0 deg, maximum phase shift 175 deg, phase shift step 10 deg. Micrographs with estimated CTF resolution of less than 3.5 Å were discarded (48 micrographs) leaving 379 micrographs for processing. Particle picking and 3D classification was performed in the same fashion as described above. 83,127 particles from the best 3D class were 3D refined producing a final reconstruction with a resolution of 2.4 Å and a B-factor of 85 $Å^2$.

## Accession codes

The reconstructed 3D density maps from the Relion particle polishing and MotionCor2 data processing workflows were deposited to the EMDataBank with accession codes EMD-3455 and EMD-3456 respectively. The dataset was deposited to the Electron Microscopy Pilot Image Archive with accession code EMPIAR-10078.

## Acknowledgements

We are grateful to Jan Schuller for kindly providing the purified 20S proteasome sample. We thank Jürgen Plitzko for his technical support.

## Additional information

### Competing interests

RD: RD is a co-inventor in US patent US9129774 B2 "Method of using a phase plate in a transmission electron microscope". WB: WB is on the Scientific Advisory Board of FEI Company. The other author declares that no competing interests exist.

### Funding

| Funder | Author |
|---|---|
| Max-Planck-Gesellschaft | Radostin Danev<br>Dimitry Tegunov<br>Wolfgang Baumeister |

The funders had no role in study design, data collection and interpretation, or the decision to submit the work for publication.

## Author contributions

RD, Conceptualization, Data curation, Formal analysis, Validation, Investigation, Visualization, Methodology, Writing—original draft, Project administration, Writing—review and editing; DT, Software, Formal analysis, Validation, Visualization, Writing—original draft, Writing—review and editing; WB, Conceptualization, Supervision, Funding acquisition, Investigation, Writing—original draft, Project administration, Writing—review and editing

## Author ORCIDs

Radostin Danev, http://orcid.org/0000-0001-6406-8993
Dimitry Tegunov, http://orcid.org/0000-0001-7019-3221

# Additional files

## Major datasets

The following datasets were generated:

| Author(s) | Year | Dataset title | Dataset URL | Database, license, and accessibility information |
|---|---|---|---|---|
| Danev R, Tegunov D, Baumeister W | 2016 | Volta phase plate with defocus cryo-EM dataset of Thermoplasma acidophilum 20S proteasome | https://www.ebi.ac.uk/pdbe/emdb/empiar/entry/10078/ | Publicly available at the Electron Microscopy Pilot Image Archive (accession no: EMPIAR-10078) |
| Danev R, Tegunov D, Baumeister W | 2016 | Thermoplasma acidophilum 20S proteasome map reconstructed with Relion 2.0 particle polishing from defocused Volta phase plate cryo-EM data | http://www.ebi.ac.uk/pdbe/entry/emdb/EMD-3455 | Publicly available at the EMDB Protein Data Bank in Europe (accession no: EMD-3455) |
| Danev R, Tegunov D, Baumeister W | 2016 | Thermoplasma acidophilum 20S proteasome map reconstructed with MotionCor2 and Relion 2.0 from defocused Volta phase plate cryo-EM data | http://www.ebi.ac.uk/pdbe/entry/emdb/EMD-3456 | Publicly available at the EMDB Protein Data Bank in Europe (accession no: EMD-3456) |

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
