## [Decision Letter]

Thank you for submitting your article "Using the Volta phase plate with defocus for cryo-EM single particle analysis" for consideration by *eLife*. Your article has been favorably evaluated by Michael Marletta (Senior Editor) and three reviewers, one of whom, Sjors HW Scheres (Reviewer #1), is a member of our Board of Reviewing Editors. The following individuals involved in review of your submission have agreed to reveal their identity: Robert M Glaeser (Reviewer #2); Raimond Ravelli (Reviewer #3).

The reviewers have discussed the reviews with one another and the Reviewing Editor has drafted this decision to help you prepare a revised submission.

Summary:

This paper is an advance on last year's paper by the same group on using the Volta phase plate (VPP) for high-resolution single-particle structure determination. Whereas last year the authors used the VPP in-focus, in this paper they collected data with a small defocus. This makes data collection easier, provides an estimate of the evolving value of the phase shift, as well as a more accurate estimate of deviations of the image defocus from the intended value. The resulting improvement in resolution, along with the decrease in the number of particles required to reach a given resolution, represents a very important advance. The resulting map is of a remarkably high resolution (up to 2.2A resolution), which proves that the VPP itself does not impose practical restrictions on currently attainable resolutions in single-particle analysis of biological samples. The paper is well written, and the results are of high interest to anyone interested in structural biology. The reviewers were therefore unanimously supportive of publication in *eLife*, provided the below is taken into account.

Essential revisions:

1) The introduction of a new software package (called Warp, which was thought to be confusing for X-ray crystallographers, who are already familiar with the Arp/Warp package) and its comparison with 2 alternative ways of processing makes this paper somewhat convoluted. The VPP results on their own, with any single one of the three processing methods, stand strong enough for publication, while the comparison distracts from the main message. In addition, the different ways of data processing differ in multiple aspects (picking algorithms, dose-weighting, CTF estimation, different particles subsets, etc.), which makes it hard to assess where the advantages of one method over the other actually lie. Probably a more rigorous comparison of the different approaches would be better suited for publication elsewhere. Therefore, the authors are encouraged to present a single result to focus this paper on the VPP results. If these results would be the ones from the Warp program, then an improved description of its methods is needed (currently the subsection “Data processing workflow with Warp” is extremely dense and rather cryptic).

2) Please comment on the fact that the amount of phase shift (as shown in Figure 1) evolves more rapidly with electron exposure in this work than was the case in the original PNAS paper in 2014, and the phase shift also rises to much higher values. The authors' thinking about why this happened, as well as their perspective about the observed variability from one area of the phase plate to another, will be really useful to other labs that currently have one of these Volta phase plate units.

3) One of the reviewers considered your resolution-dependent phase shift model, which could explain strong low-resolution components observed in images collected with a high phase shift (e.g. 0.8pi) of fundamental importance. Traditional phase plates have a cut-on frequency, here a Gaussian model was used to describe the resolution-dependent phase shift of the Volta potential. It was stated that the central phase shifting spot of the VPP can be approximated by a Gaussian function (Results and Discussion, second paragraph). However, little or no evidence was given for this statement. The shape of the potential curve of an ideal conditioned phase plate might have tails longer than what is predicted by a Gaussian curve, whereas any drifts of e.g. pivot-point misalignments might result in anisotropic potential curves. This could be better discussed. Also, the authors should provide the formulae used for Figure 2.

---

## [Author Response]

*Essential revisions:*

*1) The introduction of a new software package (called Warp, which was thought to be confusing for X-ray crystallographers, who are already familiar with the Arp/Warp package) and its comparison with 2 alternative ways of processing makes this paper somewhat convoluted. The VPP results on their own, with any single one of the three processing methods, stand strong enough for publication, while the comparison distracts from the main message. In addition, the different ways of data processing differ in multiple aspects (picking algorithms, dose-weighting, CTF estimation, different particles subsets, etc.), which makes it hard to assess where the advantages of one method over the other actually lie. Probably a more rigorous comparison of the different approaches would be better suited for publication elsewhere. Therefore, the authors are encouraged to present a single result to focus this paper on the VPP results. If these results would be the ones from the Warp program, then an improved description of its methods is needed (currently the subsection “Data processing workflow with Warp” is extremely dense and rather cryptic).*

We removed the Warp results but kept the MotionCor2 part because nowadays this is our standard workflow. Accordingly, we added a sentence: “Because of its simplicity, speed and comparable performance, nowadays we use the MotionCor2 workflow for most of our projects.” explaining this.

*2) Please comment on the fact that the amount of phase shift (as shown in Figure 1) evolves more rapidly with electron exposure in this work than was the case in the original PNAS paper in 2014, and the phase shift also rises to much higher values. The authors' thinking about why this happened, as well as their perspective about the observed variability from one area of the phase plate to another, will be really useful to other labs that currently have one of these Volta phase plate units.*

We added a paragraph (Results and Discussion, second paragraph) elaborating on the amount of phase shift and its variation between different phase plate areas.

*3) One of the reviewers considered your resolution-dependent phase shift model, which could explain strong low-resolution components observed in images collected with a high phase shift (e.g. 0.8pi) of fundamental importance. Traditional phase plates have a cut-on frequency, here a Gaussian model was used to describe the resolution-dependent phase shift of the Volta potential. It was stated that the central phase shifting spot of the VPP can be approximated by a Gaussian function (Results and Discussion, second paragraph). However, little or no evidence was given for this statement. The shape of the potential curve of an ideal conditioned phase plate might have tails longer than what is predicted by a Gaussian curve, whereas any drifts of e.g. pivot-point misalignments might result in anisotropic potential curves. This could be better discussed. Also, the authors should provide the formulae used for Figure 2.*

We added a few sentences explaining the difficulty in modeling the exact shape and profile of the VPP central spot (Results and Discussion, fourth paragraph). In that respect, the Gaussian model we used for the central spot is just an approximation of a phenomenon which is rather unpredictable in practice. The formula for the CTFs in Figure 2 was added (in the aforementioned paragraph).